# Data-Driven Short-Term Daily Operational Sea Ice Regional Forecasting

Timofey Grigoryev [1,2,*], Polina Verezemskaya [3,4], Mikhail Krinitskiy [2,3,5], Nikita Anikin [2,3], Alexander Gavrikov [3], Ilya Trofimov [1], Nikita Balabin [1], Aleksei Shpilman [6], Andrei Eremchenko [7], Sergey Gulev [3], Evgeny Burnaev [1,8] and Vladimir Vanovskiy [1,2,9]

1  Skolkovo Institute of Science and Technology, 121205 Moscow, Russia
2  Moscow Institute of Physics and Technology, 141701 Dolgoprudny, Russia
3  Shirshov Institute of Oceanology, Russian Academy of Sciences, 117997 Moscow, Russia
4  Faculty of Geography and Geoinformation Technology, National Research University Higher School of Economics, 101000 Moscow, Russia
5  Moscow Center for Fundamental and Applied Mathematics, 119234 Moscow, Russia
6  Gazprom Neft, 190000 St. Petersburg, Russia
7  Avtomatika-Service, 190000 St. Petersburg, Russia
8  Autonomous Non-Profit Organization Artificial Intelligence Research Institute (AIRI), 105064 Moscow, Russia
9  Ishlinsky Institute for Problems in Mechanics RAS, 119526 Moscow, Russia
*  Correspondence: timofey.a.grigoryev@gmail.com

**Abstract:** Global warming has made the Arctic increasingly available for marine operations and created a demand for reliable operational sea ice forecasts to increase safety. Because ocean-ice numerical models are highly computationally intensive, relatively lightweight ML-based methods may be more efficient for sea ice forecasting. Many studies have exploited different deep learning models alongside classical approaches for predicting sea ice concentration in the Arctic. However, only a few focus on daily operational forecasts and consider the real-time availability of data needed for marine operations. In this article, we aim to close this gap and investigate the performance of the U-Net model trained in two regimes for predicting sea ice for up to the next 10 days. We show that this deep learning model can outperform simple baselines by a significant margin, and we can improve the model's quality by using additional weather data and training on multiple regions to ensure its generalization abilities. As a practical outcome, we build a fast and flexible tool that produces operational sea ice forecasts in the Barents Sea, the Labrador Sea, and the Laptev Sea regions.

**Keywords:** data-driven models; short-term sea ice forecasting; deep learning; computer vision; U-Net; remote sensing; satellite imagery analysis; Arctic sea ice

## 1. Introduction

Temperature increases in the Arctic [1] are twice as high as the global mean [2–4]. According to the ERA5 reanalysis data, the annual Arctic warming trend from 1979 to 2020 is estimated to be 0.72 °C/decade [5]. Rapid Arctic warming is closely associated with an unprecedented decline of the extent of sea ice by more than 30% over the last four decades [6,7] as well as a decrease in sea ice thickness [8]. These changes allow for faster and cheaper sea routes such as the Northeast Passage [9]. Sea ice jams are some of the most critical problems in marine navigation security. Accurate operative forecasts of sea ice properties and dynamics can mitigate that problem by allowing ships to adjust their routes to avoid regions of ice accumulation. At the same time, new routes through the Arctic will cause an increase in ocean and atmospheric pollution risks, primarily due to fishing, oil/gas extraction, and transportation. For the delivery of natural gas and oil to long-distance destinations, transport by deep-sea vessels is more economical compared

to offshore pipelines [10]. To decrease ocean pollution and the carbon footprint [11,12] caused by transportation, gas/oil companies must optimize the routes [13] to make them faster and to reduce the associated ecological risks (for example, reduce the use of atomic icebreakers).

Coupled ocean-ice numerical modeling is the evident source of a reliable forecast of sea ice conditions. The newest sea ice models, such as NextSim [9,14] demonstrate fascinating results for sea ice concentration, thickness, and drift vector representations when compared to the observational data (OSI SAF SSMI-S [15], AMSR2 [16], and GloblICE dataset, available online: http://www.globice.info (accessed on 1 November 2022)). NextSim is a fully-Lagrangian finite-element model, making it tough to couple with Euler method-based ocean models. Eulerian sea ice models have been evolving for the last two decades and can reproduce some aspects of sea ice and its recent changes. However, detailed comparisons between satellite remote sensing data with Eulerian-model results reveal big differences in certain aspects of the sea ice cover, e.g., for fracture zones and small-scale dynamic processes [17,18]. It seems increasingly evident that current model physics (elastic–viscous–plastic rheology) is not suitable for reproducing these observed sea-ice deformation features [19–23] and, therefore, can not provide a reliable forecast. Furthermore, coupled ocean-ice numerical modeling requires significant computational resources.

Statistical or data-driven machine learning approaches, on the other hand, are more flexible and lightweight. That makes them popular in various research applications, even those distant from explicit computer simulations [24–26]. At the same time, when dealing with the weather and sea ice modeling, they do not need a complex physical model of processes going on in the ocean and atmosphere to work. Once trained, such a model only needs appropriate recent observations and comparatively little computational resources to make a forecast. However, the training part in this case is quite difficult for several reasons. First, most of the input data used for training (including sea ice concentration) is presented as 3d or even 4d spatiotemporal maps with a huge amount of highly correlated input channels. It has been found that usage of modern convolutional [27–31], recurrent [32,33] or attention-based [34,35] architectures can overcome difficulties associated with exploding number of trainable parameters and overfitting. Second, the model's output is expected to be a consistent SIC forecast retaining the same spatiotemporal nature, which is hard to guarantee when training on a limited amount of data. In order to overcome these difficulties, one can train a model not to predict the data itself but to compensate for the errors of simple baselines, such as climatology mean, persistence, or cell-wise linear trends. Finally, operative climate and sea ice characteristics data have their peculiarities. It is usually mosaicked, i.e., consists of several patches obtained at different times each day; thus, it should be combined and averaged daily. SIC can only be measured in the sea, leaving the land cells blank. Measurements can be based on different sources inheriting different biases, making the signal-to-noise ratio lower than expected. Furthermore, the actual changes in the sea ice condition occur in limited periods in the fall and spring, which makes more than half of the data barely usable. Considering these factors, one must be very thoughtful when designing training and testing pipelines and choose proper metrics to assess the obtained solutions adequately.

Many studies are dedicated to sea ice forecasting in the Arctic region. However, research in this field mainly focuses on climate studies rather than operative sea ice forecasts for practical use. Fully-connected multilayer perceptron (MLP) is often used either as the primary method for predicting monthly-averaged sea ice concentration [36] or as one of the benchmarks [37,38]. NSIDC Nimbus-7 SMMR and DMSP SSMI/SSMIS data are used there as SIC maps. Other approaches exploit CNN applied on patches cropped out of ice maps [38], or RF with an additional set of weather input features from ERA-Interim [39]. Deep learning methods are compared with simpler baselines in these works and are reported to perform significantly better in standard metrics such as RMSE. Refs. [40,41] are of particular interest, as they consider more advanced deep learning models that seem more suitable for sea ice forecasting. In [40], the authors consider the ConvLSTM [42] model,

which can fully make use of spatial-temporal structure of the climatological data. However, they use weather maps (predictors) from ERA-Interim and ORAS4 NEMO reanalysis data for training, thus limiting the model's applicability for operational sea ice forecasts. They evaluate the performance of ConvLSTM on a weekly-averaged and monthly-averaged scale and obtain results comparable in terms of RMSE to those of the ECMWF numerical climate model only for short lead times. The authors of [41] deal with the U-Net [30] model and train it to predict probabilities for the next 6 months for monthly-averaged SIC values in each cell to belong to each of three classes: open water, marginal ice, and packed ice. They thoroughly investigate the model's properties and compare it with SEAS5, a numerical ocean-ice model with state-of-the-art sea ice prediction skills. However, the paper does not consider possibilities for operating at the daily temporal resolution.

In this article, we focus on the operative daily sea ice forecasting and imply corresponding restrictions on the weather and sea ice data we use—it should be available for the desired regions in an appropriate resolution in the near real time for automatic downloading from a reliable source. To our knowledge, only a few papers consider this type of setting. However, all these studies either use nonoperational reanalysis data or perform experiments with one or two currently outdated machine learning methods. For example, ref. [23] demonstrates the potential of machine learning in sea ice forecasting by comparing a numerical ocean-ice model with simple CNN and cell-wise k-NN method. Unlike previous works, it focuses on short-term predictions with a length of 1–4 weeks. Ref. [43] assesses the ability of different cell-wise GRU networks equipped with feed-forward encoder and decoder to forecast SIC for up to the next 15 days. To overcome the limitations of locality in this setting, the authors incorporate global statistics in the network inputs and report significant improvement in the prediction accuracy. The authors of [44] demonstrate the superiority of ConvLSTM over CNN when forecasting SIC data in a patch-wise manner with patches of size 41 by 47 pixels. They only use NSIDC Nimbus 7 and DMSP SMMR SIC data—which is available operatively but has too low of a low resolution ($25 \times 25$ km) to be of actual use in navigation—and forecast daily-averaged SIC for the next 10 days. In [45], the authors investigate variations of relatively modern PredRNN++ [46] architecture for SIC forecasting for the next 9 days and compare it to the ConvLSTM network, demonstrating the superiority of the former. However, their model depends on ECMWF ERA5 reanalysis data, which is not available in real-time and thus limits the model's practical value.

In this study, we conducted thorough research on the prospects of machine learning in sea ice forecasting in a few regions in the Arctic: the Barents and Kara Seas (Barents), the Labrador Sea (Labrador), and the Laptev Sea (Laptev). These three regions demonstrate varying SIC interannual dynamics and allow for the investigation of the model's performance in different conditions. We deal with SIC and weather data that is available in real-time and can be used in practice to obtain operational SIC forecasts for marine navigation. A single simple yet effective classical U-Net neural architecture is chosen as such a model. It is lightweight, thus not prone to overfitting, and well suited for image-to-image tasks, such as sea ice forecasting. We treat JAXA AMSR-2 Level-3 imagery as the ground truth of sea ice concentration maps and train, validate, and test our models using this data. As a result, we not only obtain a trained U-Net model for the operational sea ice forecasts but also provide the datasets we used for benchmarks and future comparisons for the research community. All similar works test their models with different satellite data in different regions during different periods over varying baselines and usually report improvement in MAE in the range 25–50% over considered baselines. Although the comparison with them hardly makes sense, we obtain a similar daily improvement over persistence of around 25% in all three regions.

The main contributions of our work are the following:

1.  We collect JAXA AMSR-2 Level-3 SIC data and GFS analysis and forecasts data, process it, and construct three regional datasets, which can be used as benchmark tasks for future research.

2. We conduct numerous experiments on forecasting SIC maps with the U-Net model in two regimes and provide our findings on the prospect of this approach, including comparison with standard baselines, standard metric values, and model generalization ability.

3. We build a fast and reliable tool—trained on all three regions of the U-Net network that can provide operational sea ice forecasts in these Arctic regions.

4. We compare U-Net performance in forecasting in recurrent (R) and straightforward (S) regimes and highlight the strength and weaknesses of both.

## 2. Data

### 2.1. Sea Ice Data (JAXA AMSR-2 Level-3)

Plenty of sea ice concentration products are available, which cover a period from the very beginning of the satellite era to the present [47–49]. However, passive microwave sea ice concentration products are a good proxy for a large-scale ice condition assessment and assimilation in a high-resolution ocean-sea ice model. Unfortunately, they cover any specific area only once or twice a day and with data of a too low resolution (10–50 km [47]) to be of any use for the end users' requirements [50–52]. We were looking for a higher resolution satellite product to provide a forecast comparable to high-resolution (in terms of Rossby deformation radius) ocean sea ice modeling. However, even a 3 km resolution is insufficient for solving tactical planning problems [50]. We present our analysis of the sea ice conditions based on JAXA (https://earth.jaxa.jp (accessed on 1 November 2022)) as a High-Resolution Sea Ice Concentration Level-3 from Advanced Microwave Scanning Radiometer-2 (AMSR2 hereafter) from the GCOM-W satellite. AMSR2 L1B brightness temperature retrieved from antenna temperature L1A is then resampled to the pixel center points (i.e., L1R) and processed into the product (L2). The gridded L3 results from spatial and statistical temporal processing [53]. Daily sea ice concentration has been available since 2 July 2012, with a spatial resolution of 5 km on the regular grid, which is the highest resolution compared to other SIC datasets that we could find in any openly available products (SSMI, SSMIS, and other). SIC data is given in percentages (%) from 0 to 100. We used data from 2 July 2012 to 20 January 2022, i.e., 6970 days. AMSR2 L3 research product of SIC is distributed in two daily entities corresponding to the composites combined from the data acquired during ascending and descending satellite passes. In our study, we use the mean between these two daily snapshots as the ones statistically closer to ground truth compared to individual composites.

Monthly statistical distributions of SIC are presented in Figure 1 and its climatological anomalies are presented in Figure 2 (see Section 3.2 for details of its computation). Changes in sea ice are present only in about half of the months during the year. In the remaining time, the regions are fully melted (Barents and Labrador) or fully frozen (Laptev).

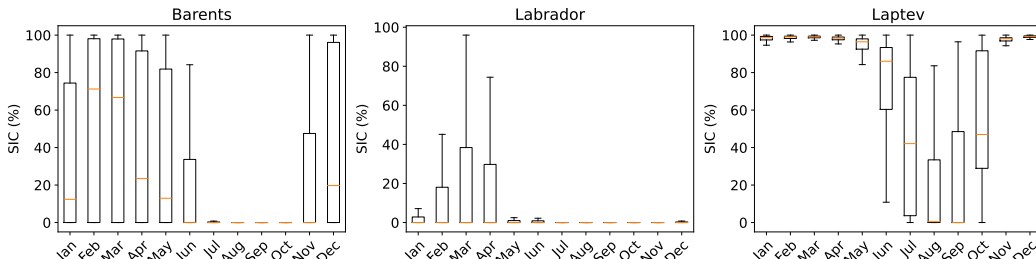

**Figure 1.** Box and whisker plots of SIC data distribution in JAXA for different months of 2021, aggregated for all the cells in each region. The box extends from the 25th percentile to the 75th percentile; whiskers extend the box by 1.5× of its length. The orange line is the median (50th percentile); outliers are omitted in order not to clutter the plot.

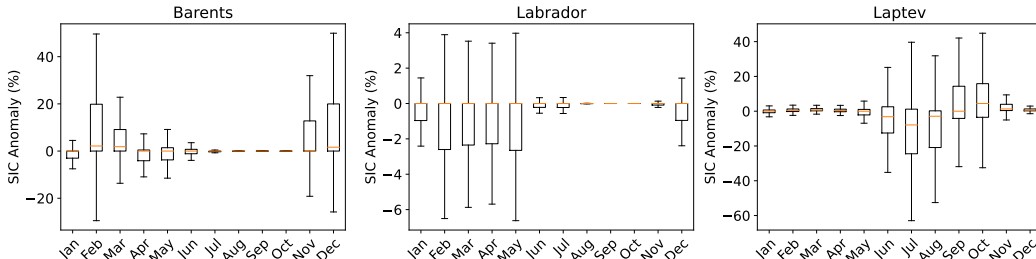

**Figure 2.** Box and whisker plots of SIC climatological anomaly distribution in JAXA for different months of 2021, aggregated for all the cells in each region. Climatological anomaly is the difference between the data and the climatology of the respective channel (see Section 3.2). The box extends from the 25th percentile to the 75th percentile; whiskers extend the box by 1.5× of its length. The orange line is the median (50th percentile); outliers are omitted in order not to clutter the plot.

### 2.2. Weather Data (GFS)

While sea ice concentration describes its condition and dynamics, there is an opportunity for potential improvement of a statistical model using additional variables correlated with sea ice dynamics. For example, surface winds influence sea ice drift, especially in shallow seas. Surface air temperature may also impact sea ice dynamics through ice melting or growth. Our study explored the potential for improving data-driven SIC forecast by extending input features with atmospheric properties such as 2 m temperature, surface pressure, and u vs. components of wind and its absolute speed.

We used NCEP operational Global Forecast System (GFS) for atmospheric and ice condition data. The GFS core is based on coupled atmospheric–ocean-ice models and provides an analysis and forecast globally at 0.25° horizontal resolution and 127 vertical levels (for atmosphere) [54]. Model forecast runs up to 16 days in advance at a 3 hourly time steps interval at 00, 06, 12, and 18 UTC daily. The output is available with no delay and a minimal number of temporal gaps, making it the best choice between the weather data sources for a reliable operative forecasting system.

### 2.3. Regions

We conduct experiments on three regions with varying SIC interannual dynamics (Figure 3). This allows us to enlarge the dataset and to adjust the model for different sea ice conditions. The Labrador Sea presents the Atlantic type of ice regime, characterized by the shortest period of pack ice in the basin (1–3 months) with mean SIC below 50% during the coldest month in a year. High interannual variability of the SIC in the Labrador Sea is caused by the sea ice fragmentation observed in the marginal ice zone (15–80%, MIZ hereafter) due to the ocean-wave–ice–atmosphere interaction. The Laptev Sea shows the typical inner-Arctic icing: SIC is above 90% spanning 7–9 months in the annual sea ice duration with strong sea ice freeze-up and slight sea ice freeze-up. The highest interannual variability is observed in summer, resulting in large open-water areas (SIC ≪ 80%). The Barents Sea and the Kara Sea regions are a mixture of these two types. The Barents Sea is an ice-free region due to the influence of intense warming from the North Atlantic Current. On the other hand, the Kara Sea is separated by the island of Novaya Zemlya and is similar to the Laptev Sea.

To prepare the dataset, we projected the regions of interest onto the new grids, which are the same except for the center points. The projection is described in Table 1. Center points are described in Table 2.

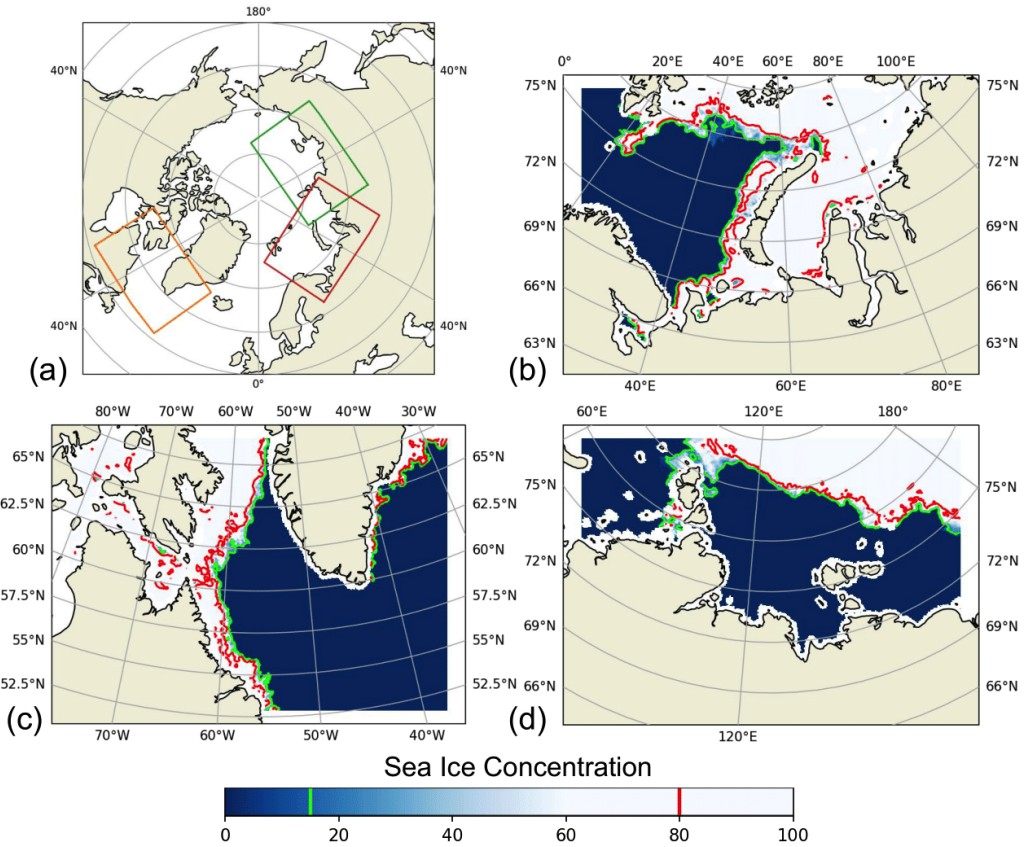

**Figure 3.** Regions definitions. (**a**) Arctic with colored boxes indicating the three regions: (**b**) the Barents and Kara Seas, (**c**) the Labrador Sea, and (**d**) the Laptev Sea. In Barents and Labrador regions, sea ice concentration is depicted for 1 April 2021, and in the Laptev region for 1 October 2021, since on 1 April 2021 the region is almost frozen. The area between the green (15%) and red (80%) isolines is the marginal ice zone for these dates.

**Table 1.** Grid parameters are the same for all the regions.

| Parameter Name | Value | Unit |
|---|---|---|
| Projection | Lambert Azimuthal Equal Area | - |
| Grid step | 5 | km |
| Grid Height | 360 | knot |
| | 1800 | km |
| Grid Width | 500 | knot |
| | 2500 | km |

**Table 2.** Central points of projections. The presentation percentage is a fraction of satellite data present in each region relative to the region's size (in cells). The lack of data is due to recognition errors and the presence of land (where SIC does not make sense).

| Region | Central Point (Lat, Lon) | Presentation Percentage |
|---|---|---|
| Barents | 73°, 57.3° | 56.82% |
| Labrador | 61°, −56° | 47.84% |
| Laptev | 76°, 125° | 51.53% |

## 3. Methods

### 3.1. Data Split

For all the regions, we used data up to the year 2019 for training, the year 2020 for validation, and the year 2021 for testing. So, for models trained just on JAXA, this means

around 7.5 years are in the training set (since mid-2012); for models trained on both JAXA and GFS, this means around 5 years are in the training set (since 2015).

### 3.2. Data Preprocessing

It is well known that artificial neural networks train more stably when fed with normalized data [55]. We compute and use the climatological anomalies instead of raw data for input channels with a strong seasonal cycle, such as GFS temperature and pressure. First, for every channel for every day in a year, we compute the climatology. It is an averaged map for that day over all the years in the training set. The averaging is performed within a window size of 3 days for further noise reduction. Then we obtain the climatological anomaly for each date by subtracting the climatology of the corresponding day from the raw data. Next, all the channels fed to the model are standardized by linear rescaling; the same is performed for every pixel (but different for different channels). The mean and variance of this transformation are computed over the training set and kept the same for validation and testing sets. Finally, the network outputs (JAXA SIC for all the cases) are rescaled by the inverse transform corresponding to the forecasted channel to be back in the desired range (0–100%).

### 3.3. Baselines

We consider three types of baselines: persistence, climatology, and trends. Persistence is a constant forecast—for any day in the future, the value of a parameter in each point is equal to that of today. The climatology baseline forecasts the historical average of a channel for that date over available observations in previous years (see Section 3.2 for details on its computation). Trends are cell-wise polynomial trends (mean, linear, quadratic, and so on) for values of a parameter for the last $D_{\text{in}}$ days. We consider trends up to cubic. Only persistence and a 3-day linear trend showed competitive results with the U-Net model. Thus, we will report only their metrics for comparison.

### 3.4. Models

We use the U-Net network [30] in all our experiments. U-Net was originally designed for image segmentation tasks. However, by not applying softmax to the last layer outputs but pixel-wise clipping them to be in the desired range, we adjust it to predict sea ice concentration in the range $[0, 1]$ instead of the logits of the classes probabilities. We exploit the same classical architecture from https://github.com/milesial/Pytorch-UNet (accessed on 1 November 2022) for all the experiments. We only adjust the amount of input and output channels to fit the chosen subsets of the variables (sea ice and weather maps).

### 3.5. Metrics and Losses

There are three classical metrics in sea ice forecasting, which differ by cell-wise statistic they aggregate: mean absolute error:

$$\text{MAE} = \frac{1}{S} \sum_{i,j \in \mathcal{A}} \left| c_{ij}^{\text{pred}} - c_{ij}^{\text{gt}} \right| dS_{ij}, \tag{1}$$

root mean square error:

$$\text{RMSE}^2 = \frac{1}{S} \sum_{i,j \in \mathcal{A}} \left( c_{ij}^{\text{pred}} - c_{ij}^{\text{gt}} \right)^2 dS_{ij}, \tag{2}$$

and integrated ice edge error [56]:

$$\text{IIEE} = \frac{1}{S} \sum_{i,j \in \mathcal{A}} \left[ \theta(c_{ij}^{\text{pred}}) \neq \theta(c_{ij}^{\text{gt}}) \right] dS_{ij}. \tag{3}$$

Here, indices $i, j$ run over all cells in active subdomain $\mathcal{A}$ of sea ice change. Our study treats all cells with present SIC data as active subdomain cells. Next, $dS_{ij}$ is the area of the respective cell, their sum

$$S = \sum_{i,j \in \mathcal{A}} dS_{ij} \tag{4}$$

is a full area of active subdomain, and $\theta(c) = [c > c_0]$ is a threshold function that binarizes SIC value $c$ with threshold $c_0$ and maps it onto one of two classes: full ice (1) or open water (0).

Our primary metric is MAE or pixel-wise $\ell_1$ loss, which is perfectly differentiable. That is why we chose to minimize it during training explicitly. For the RMSE metric, minimizing $\ell_2$ loss can be better. In our experiments, these two losses performed on par. We also considered segmentation setting with two standard classes: open water with SIC $\leq 15\%$ and packed ice with SIC $> 15\%$. For this setting, we computed the IIEE metric.

### 3.6. Augmentations

It is well known that augmentations make training more stable, prevent overfitting, and improve the generalization ability of a model [57]. We perform only geometrical transformations: random horizontal flips with a probability of 0.5; rotations on a random angle, uniformly sampled from range $[-30°, 30°]$ (with NaN padding); and translations, uniformly sampled for both vertical and horizontal shifts in the range $[-10\%, +10\%]$ of each dimension relative to map size (with NaN padding). We apply the same transformation to each channel on each input day and compare the model's output with similarly transformed target SIC maps.

### 3.7. Regimes

Classical U-Net from the box is ideally suited to capture spatial correlations in the data but not temporal correlations. In order to overcome this limitation and to avoid the complication of the architecture of the network, we consider two possible regimes to process sequential data with U-Net: a straightforward (S) and a recurrent (R) (see Figure 4).

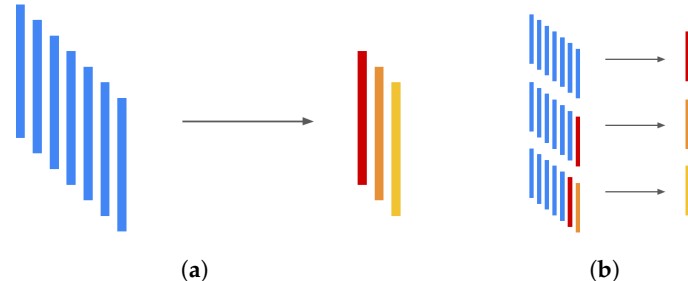

(**a**)  (**b**)

**Figure 4.** Schematic representation of the basic principles of the two U-Net regimes. Blue rods represent blocks of historical data (stacked SIC or weather maps), one for each day in the past. The leftmost is 6 days old, and the rightmost is for today (for 7-day historical input). Red, orange, and yellow rods represent model forecasts of SIC maps for the first, second, and third days in the future, respectively, (for a 3-day forecast). All the forecasts are made simultaneously in S-regime, while in the R-regime they are made one by one, and the model's inputs are updated on each step. (**a**) S-regime scheme. (**b**) R-regime scheme.

When constructing the network input, we concatenate on channels (regional maps of different scalar parameters) from the past (SIC history for the last $D_{\text{in}}$ days), channels from the future (GFS forecasts for $D_{\text{out}}$ days), and channels with auxiliary information (latitude, longitude, harmonics of the first date of the forecast, and land map). In the straightforward regime, there are $D_{\text{out}}$ output channels—one for SIC of each forecast day. In the recurrent mode, we only predict the SIC for the next day and use it iteratively as an input along with other channels in a recurrent fashion for $D_{\text{out}}$ times to construct a forecast for $D_{\text{out}}$ days. However, training that way seems to be subject to the same limitations as any RNN [58]

and does not go well. Therefore, first we pretrain U-Net to make a forecast for $D_{\text{out}} = 1$ day and then finetune for $D_{\text{out}} > 1$ in the described fashion. This approach is inspired by ideas of curriculum learning [59] and improves the results significantly compared with no finetuning or no pretraining and allows one to make stable forecasts further into the future.

### 3.8. Implementation

We implemented a training and testing pipeline in Python using the popular machine learning framework PyTorch. We used an Adam optimizer with a learning rate $10^{-4}$, learning rate decay rate $10^{-2}$, and batch sizes of 16 in the S-regime and 8 in the R-regime. We conducted each experiment on a single NVIDIA A100 40 GB GPU, requiring from 1 to 12 h for training depending on the number of input channels, number of involved regions, and the training regime.

## 4. Results

As was mentioned in Section 3.7, we consider two regimes for time series forecasting using the U-Net backbone: straightforward (S), when the number of output channels is set equal to the number of output days, and recurrent (R), when the number of output channels is set to 1 and the model is iteratively applied for the number of output days times in recurrent fashion. In this section, we will provide the results of the conducted experiments for both regimes in different settings and compare them. We conduct experiments on all three regions (Barents, Labrador, and Laptev) and present the results as well. The scales of the same diagrams and plots for different regions may vary depending on the specifics of geographical features and sea ice dynamics in each region.

### 4.1. Inputs Configuration

There is a vast number of combinations of possible inputs for the model. JAXA data has just one channel—SIC—but we cannot vary only the number of previous days we stack for the model but also the channel's preprocessing. One can compute climatology and the climatological anomaly (see Section 3.2) and pass it as well, just for the last day or for all of the input days. The same goes for GFS data, from which we could use five channels (temperature, pressure, u, and vs. components of wind and its module) and four forecasts with a different lead time for each day. On top of that, one can suggest several general channels that can be useful. Firstly, one should consider the harmonics of the current day phase in the year. That is $\cos \varphi$ and $\sin \varphi$ for $\varphi = 2\pi \frac{D}{D_Y}$, where $D$ is a number of a current day from the start of the current year and $D_Y \approx 365.2425$ is the average number of days in a year. It is natural to assume the dynamics of sea ice are different in different seasons, and that input will allow a model to capture these dependencies. Secondly, the binary segmentation map of sea and land may be useful. The region of interest is entirely in the sea, so it can be useful for the model to treat weather data from the land differently if it falls in the perception core of the convolutions. Thirdly, one can use a map of areas of the grid cells. In our case, all the grid cells are almost the same area due to the choice of the projection type (see Section 2.3). Finally, the grid cells' longitudes and latitudes might also be useful. They can be used for a model trained to perform on one region to better fit parts of it with different climatological properties. On the other hand, they can harm the model's generalization ability and drop its performance in other regions if the values are out of the neural network domain.

It is worth mentioning that stacking all the available channels into the model's input is not the best solution since many of them are highly correlated and some of them may have no relevant signal for the model. We considered the nature of each channel and conducted many experiments. This allowed us to choose a single configuration of inputs for both regimes that includes all the available and useful channels. The configuration is described in Table 3. For the experiments with no GFS data, we omitted all the channels with source GFS.

**Table 3.** Chosen configuration of the inputs for the experiments. "Data" in preprocessing means that no preprocessing except standardizing was performed. "Past" time interval means stacking all the specified maps for all the days in the past, including the last observable day ("Today"). "Future"—stacking all the forecasts for the output days (3 for S-regime and 1 for R-regime). In R-regime, the corresponding forecasts from the last observable day replace data and forecasts of the coming days, so that no yet unobserved data or forecasts leak to the model from the future.

| Source | Channel | Preprocessing | Time Interval |
|---|---|---|---|
| **JAXA** | **SIC** | **Data** | **Past** |
| GFS | Temperature | Climatology | Today |
| | Temperature | Clim. Anomaly | Today |
| | Temperature | Clim. Anomaly | Future |
| | Pressure | Climatology | Today |
| | Pressure | Clim. Anomaly | Today |
| | Pressure | Clim. Anomaly | Future |
| | Wind (u) | Data | Today |
| | Wind (u) | Data | Future |
| | Wind (v) | Data | Today |
| | Wind (v) | Data | Future |
| | Wind (module) | Data | Today |
| | Wind (module) | Data | Future |
| General | Date (cos) | Data | Today |
| | Date (sin) | Data | Today |
| | Land | Data | Today |

### 4.2. Predicting Differences with a Baseline

The SIC map by itself is quite a complex image. Although U-Net architecture is well-suited for predicting local changes on an image, it may still struggle to reproduce the whole input, which may be similar to the desired output with some local changes or require additional training time. In order to alleviate the problem for the model and to accelerate its training, we do not make it predict the SIC data and instead use it to predict the differences with a baseline. Since the persistence baseline performed best, we used it as the base $B$ and computed the model's forecast $F$ by using the formula:

$$F = B + \alpha M. \tag{5}$$

Here, $M$ is the backbone's (U-Net) output, and we chose $\alpha = 0.1$ so that backbone outputs $M$ have approximate unit variance (which is the best for the default weights initialization [60]). We investigate the effect of this decision when we do ablations in Section 4.6.

### 4.3. Pretraining in R-Regime

In R-regime, we recurrently make the next-day SIC forecast by using all the previous predictions as inputs (see Section 3.7). The gradients flow through the backbone for $D_{out}$ times in this setting. We discovered that training from scratch in this regime proceeds very slowly and falls to nonoptimal solutions that perform much worse than models trained in the S-regime. The reason may be that the R-regime is much more sensitive to proper initialization. To solve this problem, we divide the whole training set into two parts:

1. Pretraining the model in S-regime with $D_{out} = 1$;
2. Initializing the model with the pretrained checkpoint and tuning it in R-regime for $D_{out}$ days.

The pretraining is conducted as usual for 100 epochs, and the tuning is only conducted for 20 epochs, which we found to be sufficient.

*4.4. 3 Days Ahead Forecast*

In this subsection we will describe results of the experiments conducted for $D_{in} = 7$ input days and $D_{out} = 3$ output days for the general model (trained on all three regions) with GFS data. This number of input days was chosen from these theoretical considerations: the model should require appropriate computational and memory resources for training yet be able to catch all the necessary trends and dynamics in the data. The number of output days is thought to be sufficient to investigate the model's forecast properties and compare the model's abilities in different settings. On the performance of the model with different numbers of input days see Section 4.6 and for the longer forecasts see Section 4.5. All the metric values for our best models and baselines are collected in Table 4.

**Table 4.** JAXA SIC metrics averaged over 3 forecast days and over 2021 for baselines and our best U-Net configurations (general with GFS). IIEE is computed for SIC classes with a 15% binarization threshold. For the models, we report mean and unbiased std of 3 independent runs with random seeds 0, 1, and 2. The best mean value in the each row is printed in bold.

| Region | Metric | Linear Trend | Persistence | U-Net (S) | U-Net (R) |
|---|---|---|---|---|---|
| | IIEE | 2.96 | 2.46 | 1.48 ± 0.02 | **1.41** ± 0.009 |
| Barents | MAE | 3.25 | 2.67 | 1.78 ± 0.01 | **1.73** ± 0.004 |
| | RMSE | 9.8 | 8.44 | 5.68 ± 0.05 | **5.51** ± 0.05 |
| | IIEE | 1.82 | 1.54 | 0.905 ± 0.004 | **0.871** ± 0.01 |
| Labrador | MAE | 1.66 | 1.41 | 0.966 ± 0.003 | **0.939** ± 0.009 |
| | RMSE | 6.59 | 6.02 | 3.96 ± 0.03 | **3.88** ± 0.05 |
| | IIEE | 2.03 | 1.7 | 1.11 ± 0.03 | **1.05** ± 0.02 |
| Laptev | MAE | 3.7 | 3.03 | 2.22 ± 0.02 | **2.16** ± 0.007 |
| | RMSE | 8.87 | 7.61 | 5.1 ± 0.06 | **4.98** ± 0.05 |

Since the forecasts with longer lead times (that are further in the future) are more challenging due to accumulating uncertainty, the error rate should increase as the number of lead-time days increases. This dependence is depicted in Figure 5. U-Net in both regimes outperforms both baselines by a significant margin. Interestingly, the linear trend performs noticeably worse than persistence, which we associate with the high nonlinearity of the SIC dynamics in each cell along with measurement errors. U-Net (R) performs slightly better than U-Net (S) in all the cases, and the difference tends to increase with an increase in the forecast lead time. That might indicate that the U-Net (R) model is more suitable for longer forecasts as it is trained in a way that mitigates its errors when receiving the outputs as the next day's inputs. Examples of forecasts for fixed dates with 1, 2, or 3 lead-time days with the best configuration of U-Net (R) are presented in Figure 6. The green lines depicted in Figure 6 contour the MIZ and when overlaid with the error (red–blue) it shows that, like in numerical models, the most considerable discrepancies are contained inside it. One can see that most of the errors are accumulated near the edge of the ice, where most of the daily sea ice changes take place.

More informative are Figure 7a,b, where absolute improvements of the model over persistence are presented separately for each month and color-coded. The similarity between Barents and Labrador regions, where the most improvement is obtained during winter and spring, and their distinction from the Laptev region, where the most improvement is obtained during summer and fall, are apparent.

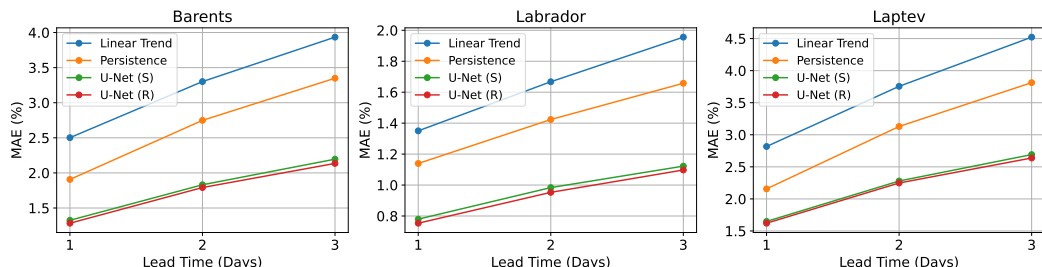

**Figure 5.** Dependence of JAXA SIC MAE (lower is better) on the different forecast horizons (in days) for all three regions. MAE is averaged over the whole 2021 year. The linear trend is computed cell-wise over 3 previous days; U-Net (S) and U-Net (R) are trained on all three regions merged and shuffled, with 7 days history (past) and best inputs configuration, as presented in Table 3.

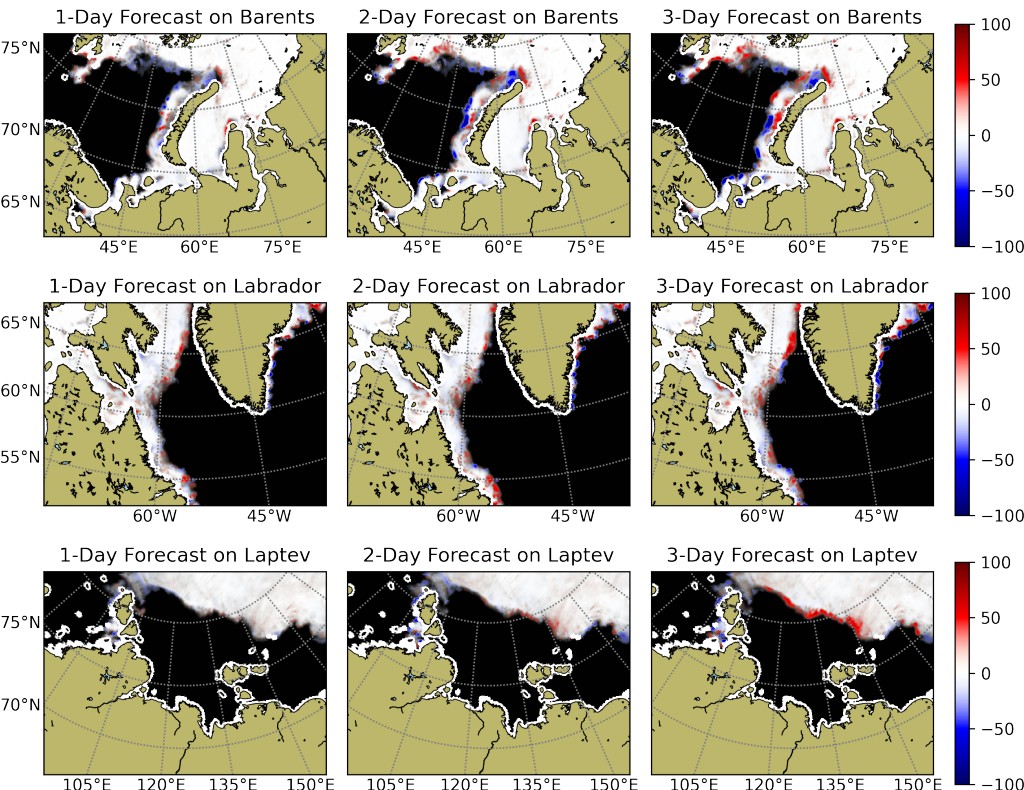

**Figure 6.** U-Net (R) best configuration forecasts examples for fixed dates and varying forecast lead time. For Barents and Labrador, the date is 1 April 2021, and for Laptev the date is 1 October 2021, since on 1 April 2021 the region is almost frozen. Black–white color map shows the values of JAXA SIC, and the semitransparent red–blue color map shows the difference between model predictions and actual values of SIC. MIZ is not shown for readability of the maps, but it is located predominantly along the sea ice edge and occupies approximately 10% of the respective sea area for the selected dates. For more forecast visualizations, please refer to the supplementary material.

As was mentioned earlier, U-Net (R) performs slightly better than U-Net (S). This improvement is shown in Figure 7c. U-Net (R) enhances the solution mostly during the generally challenging seasons: autumn/spring (due to icing/melting) like it was with the persistence baseline. Changing the ice regime from winter to summer is accompanied by enlarging the marginal ice zone—the area with the highest sea ice activity. This activity can be evaluated using statistical analysis of the distribution of cells' SIC climatological anomalies for each month, such as those depicted in Figures 1 and 2. Another way to estimate the impact of MIZ area on the solution quality is to compute the relative area (%) of the marginal ice zone in the ocean domain (orange plot in Figure 8). In Figure 8, we see

that the correlation between the MAE and MIZ relative areas almost equals 1. The high correlation means that even not rheology-dependent ML models face the same problem as numerical models [61].

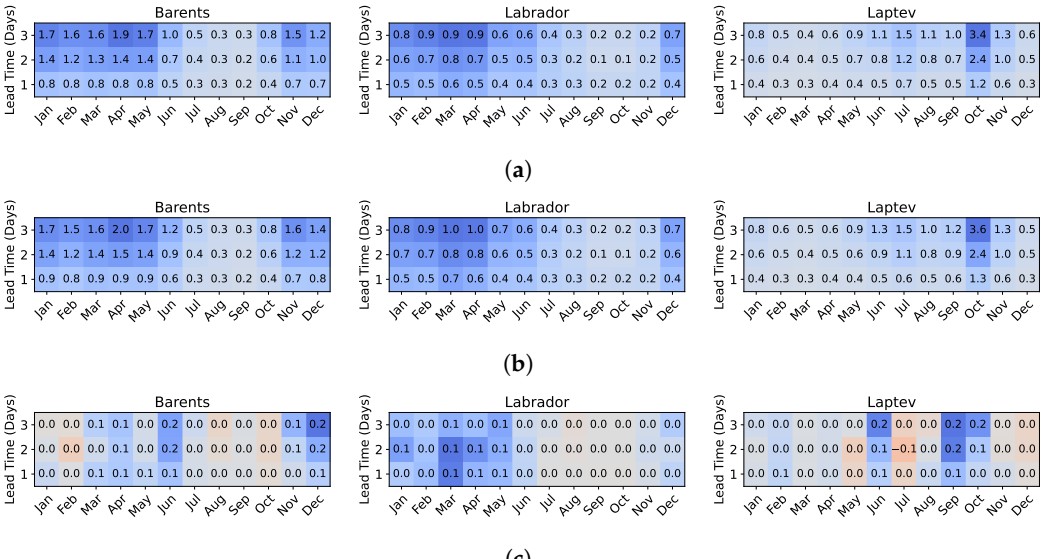

**Figure 7.** Improvement of JAXA SIC MAE (in percentage points, higher is better) for different models compared to the other models or baselines for different months of 2021 and days of the forecast. The matrix for each region and subfigure is color-coded independently in a red-blue scheme to provide a better perception of the relative improvement for each month and forecast day in the corresponding region. The numbers in each cell are rounded and should be used to estimate the overall improvement scale in the regions. Improvement is computed in absolute percentage points and tends to be higher for months with active sea ice change. (**a**) General U-Net (S) with GFS over persistence. (**b**) General U-Net (R) with GFS over persistence. (**c**) General U-Net (R) with GFS over general U-Net (S) with GFS.

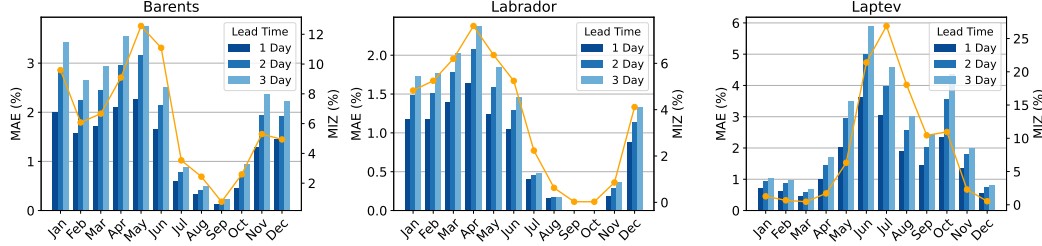

**Figure 8.** General U-Net (R) with GFS: distribution of JAXA SIC MAE (lower is better) for different months of 2021 and days of the forecast. The orange line shows the marginal ice zone area relative to the region's sea area.

All the results above are given for the general models. That means that both U-Net (S) and U-Net (R) were trained on three regional datasets merged into one and shuffled. One can expect that more diverse data will be helpful for the model to perform better. To demonstrate this, we conducted experiments and compared the general model with all three regional ones in both S- and R-regimes. The performances of all the settings are shown in Figure 9. The most boost is achieved by including GFS fields in the inputs; switching between general and regional settings helps most in the Laptev region, does not change in the Barents region, and gives a nonsignificant decrease in performance in the Labrador region. The change when switching from regional models to the general model is shown in Figure 10 for U-Net (R) (for U-Net (S) it looks almost the same). In Figure 11, regional models are also tested on the other two unseen regions and compared to the general model. In both cases, the upper 3 × 3 square diagonal is dark blue because the models are tested on their native regions, and the bottom row depicts the general model's results. We prefer the

general model to regional models because of its universality, and we expect it to generalize better when analyzing previously unseen regions.

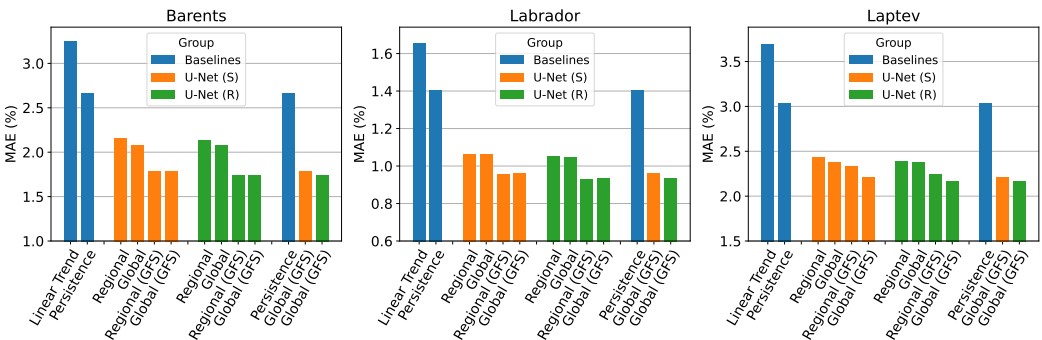

**Figure 9.** JAXA SIC MAE averaged over 3 days of forecast and over 2021 (lower is better) for baselines and different configurations of U-Net. Colors denote model groups (baselines, U-Net (S), and U-Net (R)), configuration of each model within a group is specified below each bar. For U-Net regional and general configurations were trained with or without GFS. The best configurations from each group are presented separately on the right of each region plot for comparison.

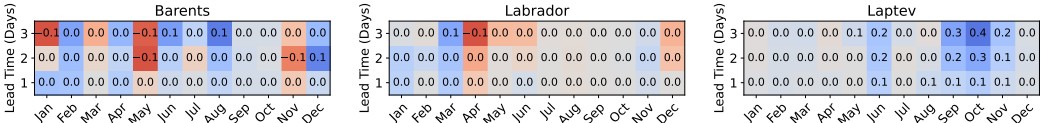

**Figure 10.** Improvement of JAXA SIC MAE (in percentage points, higher is better) for general U-Net (R) with GFS over regional U-Net (R) with GFS for different months of 2021 and different days of the forecast. The matrix for each region and subfigure is color-coded independently in a red-blue scheme to provide a better perception of the relative improvement for each month and forecast day in the corresponding region. The numbers in each cell are rounded and should be used to estimate the overall improvement scale in the regions. Improvement is computed in absolute percentage points and tends to be higher for months with active sea ice change.

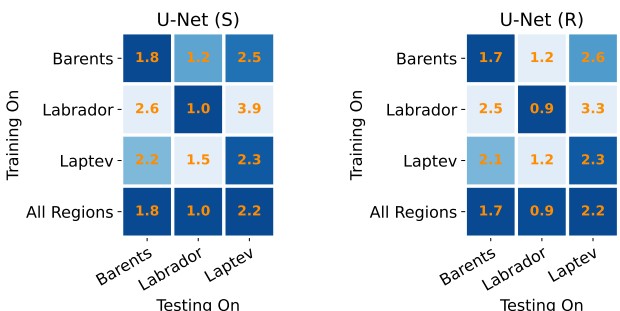

**Figure 11.** JAXA SIC MAE averaged over 3 days of forecast and over 2021 (lower is better) for different train-test region configurations. The models trained in any region are expected to perform best in this region, and the general model (trained in all three regions) is expected to perform well in all of them. Each column of both matrices is color-coded independently in a white-blue scheme so that lesser MAE values correspond to a darker shade. Independent coloring provides a better perception of the best training setting for each testing region.

### 4.5. 10 Days Ahead Forecast

We also trained U-Net (S) to make a forecast for $D_{\text{out}} = 10$ days and tested U-Net (R) trained with $D_{\text{out}} = 3$ for 10 output days. We did not train U-Net (R) with $D_{\text{out}} = 10$ days because of memory and computational requirements, which, in this case, were too big. We also trained the U-Net (R) model without GFS channels since we only had GFS forecasts for the next 3 days. The results are depicted in Figure 12. To better understand the influence of the presence of GFS channels in the inputs, we also conducted experiments for U-Net (S)

with GFS data and computed relative improvement over persistence for all three settings. The results are presented in Figure 13. One can see an improvement of 5–15% for the first 3–4 days for the U-Net (S) with the GFS setting in the Barents and Laptev regions compared to U-Net (S) without the GFS setting. For the Labrador region, the improvement is smaller but persists for all 10 days of forecast, which is the same for the Barents region. U-Net (R) without GFS generally performs better for the first half of the forecast days than U-Net (S) without GFS. Its performance deteriorates as the forecast lead time increases. This fact is general knowledge about RNNs, whose performance usually deteriorates when the depth of recurrence increases.

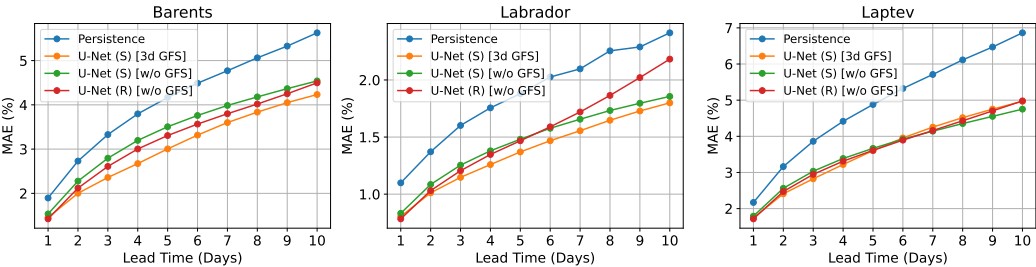

**Figure 12.** Dependence of JAXA SIC MAE (lower is better) on the number of forecast day in the future (lead time of the forecast) for all three regions. MAE is averaged over the whole year 2021. U-Net (S) and U-Net (R) are trained on all three regions merged and shuffled with 7 days history (past) and best inputs configuration, as presented in Table 3.

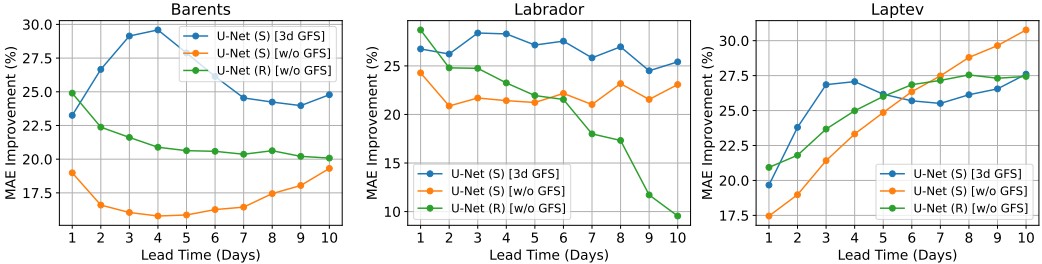

**Figure 13.** Dependence of JAXA SIC MAE relative improvement over persistence (higher is better) on the different forecast horizons (in days) for all three regions. MAE is averaged over the whole year 2021. U-Net (S) and U-Net (R) are trained on all three regions merged and shuffled with 7 days history (past) and best inputs configuration, as presented in Table 3.

### 4.6. Ablation Studies

In Section 3.6, we introduced the set of augmentations that we used. In Section 4.2, we described how we did not predict raw SIC data but rather the difference with the persistence baseline to alleviate the problem for U-Net. In Figure 14, we demonstrate the influence of both these factors on the model's accuracy for our best configuration (general U-Net (R) with GFS). They boost MAE in the Barents and Laptev regions and do not make any significant difference in the Labrador region.

We also investigated the influence of including the GFS channels in the model and dissected the improvement over months. The results for U-Net (R) are presented in Figure 15 and are similar to those for U-Net (S). In both cases, these channels proved to be very useful for the model, especially during the months with an active change of sea ice.

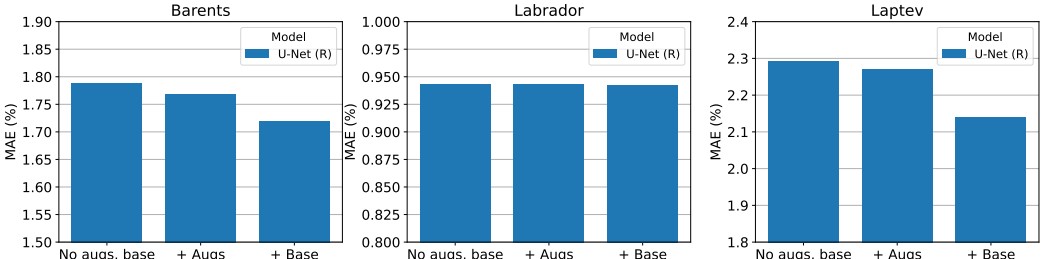

**Figure 14.** JAXA SIC MAE averaged over 3 forecast days and over all of 2021 for different ablations of general U-Net (R) configuration with GFS. From left to right on each diagram, we first turn on augmentations and then add persistence base to the predictions (as in Equation (5)) and report the model's metric.

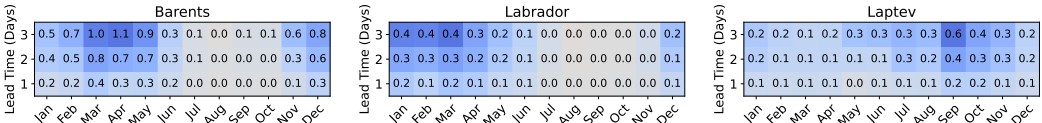

**Figure 15.** Improvement of JAXA SIC MAE (in percentage points, higher is better) for general U-Net (R) with GFS over general U-Net (R) without GFS for different months of 2021 and different forecast horizons (in days). The matrix for each region and subfigure is color-coded independently in a red-blue scheme to provide a better perception of the relative improvement for each month and forecast day in the corresponding region. The numbers in each cell are rounded and should be used to estimate the overall improvement scale in the regions. Improvement is computed in absolute percentage points and tends to be higher for months with active sea ice change.

## 5. Discussion

In short-term predictions (for 3 days), U-Net (R) slightly outperformed U-Net (S), and both of them outperformed baselines by a significant margin (Table 4), thus demonstrating high prospects of machine learning methods for sea ice forecasting. For longer lead times (10 days ahead forecasts), U-Net (R) quality deteriorates as anticipated, and it usually yields to U-Net (S) starting from the second half of the forecast (Figure 13). Extra weather forecast channels improve forecasts quality noticeably for the days when the weather forecast is provided and to some extent for subsequent days (Figure 13). The marginal ice zone stands as the most challenging part of a region, and the months of most active ice change stand as the most challenging part of a year for forecasting (Figures 6 and 8). Finally, training not only on the region of interest but also on more diverse data improves the overall performance of the model and its generalization abilities (Figure 11).

We utilized U-Net architecture for our experiments, and it performed well. It is lightweight, thus not prone to overfitting, yet well suited for image-to-image tasks, such as sea ice forecasting. However, there are a few more specialized neural network architectures used mainly in video prediction tasks (PredRNNs [46,62,63], E3D-LSTM [64], CrevNet [65]) that are closely related to sea ice forecasting. One possible direction for future work can be embedding these backbones into the pipeline and comparing their performance. Another important extension is to compare the performance of numerical ocean-ice models, such as GLORYS12-V1, TOPAZ4, or SODA3.3.1, with our developed data-driven approach. One should also consider exploring more advanced approaches to measure the quality of the forecasts in terms of their value for marine operations. Ref. [61] showed that the ten most modern ocean reanalyses systematically underestimate the area of MIZ during spring and autumn even with data assimilation. Finally, one can try to fuse these approaches and train statistical models to compensate for errors of the numerical models or use them in any other more intricate way. These may include more flexible sparse data assimilation (e.g., from buoys), incorporating physics into data-driven models via NeuralODE approaches [66], or exploiting advances in other machine learning fields. For example, one can try various optimal transport methods [67,68] for the prediction of the sea ice drift.

Adding extra data, particularly data that represents types of sea ice and sea ice thickness, may also significantly boost the forecasting quality. The main difficulty here may be to locate reliable and complete sources of these operative data.

## 6. Conclusions

Data-driven models based on machine learning are gaining popularity as fast and robust alternatives for numerical ocean-ice models in short-range weather forecasting. We investigated their efficiency in sea ice forecasting in several Arctic regions. First, we collected JAXA AMSR-2 Level-3 SIC data and GFS analysis and forecasts data, processed it, and then constructed three regional datasets, which can be used as benchmark tasks in future research studies. Second, we conducted numerous experiments on forecasting SIC maps with the U-Net model in two regimes and provided our findings on the prospect of this approach, including comparison with standard baselines, standard metric values, and model generalization ability. That allowed us to build a fast and reliable tool—trained on all three regions U-Net network—that can provide operational sea ice forecasts in any Arctic region. Finally, we compared U-Net forecasting performance in recurrent (R) and straightforward (S) regimes and highlighted the strengths and weaknesses of both these regimes.

**Supplementary Materials:** The following supporting information can be downloaded at: https://www.mdpi.com/article/10.3390/rs14225837/s1, https://disk.yandex.ru/d/n3PaW0p04GPiwg, Video S1: barents-1d-forecasts.mp4, Video S2: barents-2d-forecasts.mp4, Video S3: barents-3d-forecasts.mp4, Video S4: labrador-1d-forecasts.mp4, Video S5: labrador-2d-forecasts.mp4, Video S6: labrador-3d-forecasts.mp4, Video S7: laptev-1d-forecasts.mp4, Video S8: laptev-2d-forecasts.mp4, and Video S9: laptev-3d-forecasts.mp4.

**Author Contributions:** Conceptualization, V.V., E.B., P.V. and S.G.; methodology, V.V., T.G., M.K. and N.B.; software, T.G., M.K. and N.A.; validation, T.G., P.V., A.S. and A.E.; formal analysis, T.G., M.K., N.A. and I.T.; investigation, T.G.; resources, E.B.; data curation, M.K., N.A., P.V. and A.G.; writing—original draft preparation, T.G., P.V., M.K., N.A. and A.G.; writing—review and editing, V.V., E.B., N.B., I.T., A.S., A.E. and S.G.; visualization, T.G. and N.A.; supervision, V.V., M.K., A.S. and A.E.; project administration, V.V. and E.B.; funding acquisition, E.B. and V.V. All authors have read and agreed to the published version of the manuscript.

**Funding:** The work was supported by the Analytical center under the RF Government (subsidy agreement 000000D730321P5Q0002, Grant No. 70-2021-00145 02.11.2021).

**Data Availability Statement:** Interpolated and preprocessed data for all three regions (Barents, Labrador, and Laptev) used in this study are openly available at https://disk.yandex.ru/d/5Qc_OhbU7NYQew (accessed on 1 November 2022). See README.md there for the details.

**Acknowledgments:** We gratefully appreciate insightful discussions with experts from Gazprom Neft and Avtomatika Service and thank the Association "Artificial intelligence in industry" for providing a platform for such discussions.

**Conflicts of Interest:** The authors declare no conflict of interest. The funders had no role in the design of the study; in the collection, analyses, or interpretation of data; in the writing of the manuscript; or in the decision to publish the results.

## Abbreviations

The following abbreviations are used in this manuscript:

| | |
|---|---|
| AMSR-2 | Advanced Microwave Scanning Radiometer 2 |
| CNN | Convolutional neural network |
| CPU | Central processing unit |
| ConvLSTM | Convolutional LSTM |
| DMSP | Defense Meteorological Satellite Program |
| ECMWF | European Centre for Medium-Range Weather Forecasts |
| ERA | ECMWF re-analysis dataset |

| | |
|---|---|
| GFS | Global Forecast System |
| GPU | Graphics processing unit |
| GRU | Gated recurrent unit |
| IIEE | Integrated ice edge error |
| JAXA | Japan Aerospace Exploration Agency |
| k-NN | k nearest neighbors |
| LSTM | Long short-term memory |
| MAE | Mean absolute error |
| MIZ | Marginal ice zone |
| MLP | Multilayer perceptron |
| NEMO | Nucleus for European Modelling of the Ocean |
| NOAA | National Oceanic and Atmospheric Administration |
| NSIDC | National Snow and Ice Data Center |
| ORAS4 | Ocean Reanalysis System 4 |
| RF | Random Forest |
| RMSE | Root mean square error |
| RNN | Recurrent neural network |
| SAR | Synthetic-aperture radar |
| SIC | Sea ice concentration |
| SMMR | Scanning multichannel microwave radiometer |
| SSMI | Special Sensor Microwave/Imager |
| SSMIS | Special Sensor Microwave Imager/Sounder |

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
