# Peer review of "Data-Driven Short-Term Daily Operational Sea Ice Regional Forecasting"

_remotesensing, doi:10.3390/rs14225837_

Round 1
Reviewer 1 Report
The authors have done a very good job in presenting novel approach to using similar methods for sea ice forecasting relevant for operations and services. However, a great deal of work has been done with researchers on sea ice models such as CICE, NextSim and others, that have identified the need for higher spatial resolution input data so that the forecasts are suitable for maritime users working on tactical and short-term scales. The authors should review the metrics and in this paper and incorporate additional updated approaches that and the use of validation data that is on the scale of users such as synthetic aperture radar, optical (where available) or visible infrared and ice charts. The following are additional comments:
P1 L15-17. Rephrase grammar.
P2 L33. "NextSIM" should be "NextSim"
P2 L35-36. Only reference #15 has been used indirectly, not the others.
P2 L41-43. Probably trending towards no, please see Fritzner, Graversen and Christensen, 2020 and Fritzner et al. 2019. Revise
P2 L59-60. Long winded way of saying "mosaicked"
P2 L69. Expand MLP rather than in the glossary at the end.
P3 L116 Clarify the limitations of PMW here. Should not be referred to as ground truth, rather a validation dataset
P4 L140. "Nowadays" should be changed to "the present."
P4 L141-145. The term high-resolution should be clarified that the 5km spatial resoulution is not considered to be high resolution in terms of sea ice forecasting for tactical applications. Sea ice forecasts to be useful for sea ice services are requested to be less than 1km spatial resolution (Hunke et al., 2020, Blockley et al., 2020, Wagner et al. 2020 etc..). Clearly state this and reference previous relevant work related to this topic, as there has been a lot in the last 4 years).
P4 L148. Is this supposed to mean CRYOSAT2? CRYOSAT and CRYOSAT2 has a higher resolution than AMSR2 as a radar altimeter at 250m. Also, CRYOSAT is not a SIC dataset.
P6. F3. This figure is confusing because it does not specify the dates in these areas. Also, there is an area specificially showing the concentration in the Laptev Sea when it seems that the point of the figure is to show that the MIZ is between the red and green concentration isolines. Based on this figure, and given many times throughout the year, the area (specified) between the red and green lines should not be regarded as the marginal ice zone (MIZ) because compact and thicker sea ice forms in many places in between these areas. This is too large to be considered a marginal ice zone. We would also expect the MIZ in the Laptev Sea some parts of the year. Revise this figure to be more clear.
P12 F6. This figure needs to be revised. It is difficult to asssess results on postage stamp-sized maps even with zoom-in on PDF. The green lines are observations, not forecasts and distract. The colors used will be an issue for color-blind readers so consider a different color scheme. Last, the significant areas of diff[erence] - red and blue are diffficult to see.
P12 F7, 8 and 9. This is really confusing that these are broken up by U-Net S and R, followed by text in between and yet Figure 9 is supposed to link Figure 7 and 8? This should either be in one figure or provide a different visualization of the R, S and difference of the two, because it is very difficult to understand the overall message.
P17 L436 The main drawback as an operational tool is PMW. The metrics used in this paper does not seem appropriate to understand how the model performs in an operational or service perspective. The presentation of regional statistics fails to explore differences in MIZ that would be of interest for ice operations or end-users by using bulk statistics. A revision and zoomed-in Fig[ure] 6 would be nice. Additionally, the authors should seek out literature that has already been done to address some of these issues such as, the use of quantitative statistics using IIEE (Goessling, H. F., Tietsche, S., Day, J. J., Hawkins, E., and Jung, T. (2016), Predictability of the Arctic sea ice edge, Geophys. Res. Lett., 43, 1642– 1650, doi:10.1002/2015GL067232 and others that have published on this topic since 2016) would be of interest.
Reviewer 2 Report
The manuscript is devoted to currently the problem of forecasting the concentration of sea ice in the Arctic region. The authors carried out quite serious U-NET applications for the short-term modeling of the concentration of sea ice (3 days in advance) in the study region (in the Barents, Karsk Seas, the Labrador Sea and the Laptev Sea) from a technical point of view. The article obtained conclusions relatively better U-NET configuration to solve the task that gives the lowest forecast error in the 3.5 metrics specified in paragraph. Numerical experiments were also conducted to forecast ice concentration for 10 days. Conclusions have been obtained about the importance of using GFS data. However, the article requires some refinement, and here is a list of my comments:
1. In the introduction, it should be noted that the data of remote sensing from various satellites often contradict each other and have errors (sometimes very significant). When using classic models, as a rule, they seek to explore the sensitivity of the solution to errors in the source data. Are the authors know such works for models based on neural networks?
2. On page 3, lines 89-90 - from the context it is not very clear what kind of restrictions on data on the atmosphere and ice are discussed.
3. In paragraph 2.1, there is not a description of which pre -processing AMSR2 is undergoing when creating the AMSR2 L3 product, and what is their fundamental difference from AMSR2 L4. It is also necessary to add links to the relevant documentation and comment on possible sources of error in the data. The reviewer considers this necessary, since the neural network is studying precisely on these data, therefore, when constructing the forecast, errors in the data will be reproduced. So, for example, if during processing data, large errors are allowed in the area of ​​the water and ice, in the construction of the forecast in this zone, a large error is also possible.
4. P. 5, line 171 - indicating the data format is an extra technical detail that goes beyond the scope of the scientific article.
5. P. 11, paragraph 325-333. What physical prerequisites or reasoning underlie the fact that the forecast is built for 3 days?
6. It would be interesting to look at the modeling results when using other, more complex ML models, for example, Convlstm or Flownet.
7. According to the reviewer, the most interesting study in the future is to compare the calculation results using ML models and classic models, namely a comparison of calculation results in regions with different types of ice, and comparing the machine time used in calculations. So, in the end, the reader will be able to understand in which cases it is advisable to use classic models, and in which ML models. Moreover, it would be interesting to see what results it can be achieved using the data on the surface currents from reanalysis as additional channels for various hydraulic models.
8. There were typos in punctuation, for example:
a. p. 2, line 52 - “Found, that” a comma is not needed.
Reviewer 3 Report
This paper developed a machine learning based on data-driven models to forecast sea ice concentration. It is an interesting and meaningful topic which supple Arctic Sea ice concentration data.
About data:
1. The input features for the model include the atmospheric and ice condition data such as the 2-meter temperature, surface pressure and so on. The certainties for the input data should be introduced.
About methods:
1. The detailed process of algorithm to predict sea ice concentration can be presented in the map which will be clearer.
About conclusion and discussion
1. About the input data, the spatial resolution of AMSR2 sea ice concentration is different from the atmospheric data. If the author deal with the different resolution problem and what is the spatial resolution of sea ice concentration from the forecasting model?
2. The sea ice concentration from the model should be compared with the SSM/I and SSM/IS which can be used to evaluate the reliability of the model output.
3. The certainties of output for the forecasting model should be discussed.
